# Reconstruction of the Hip in Multiple Hereditary Exostoses

**DOI:** 10.3390/children8060490

**Published:** 2021-06-08

**Authors:** Dong Hoon Lee, Dror Paley

**Affiliations:** 1Donghoon Advanced Lengthening Reconstruction Institute, Superstar tower 3-5F 10, Wiryeseoil-ro, Sujeong-gu, Seongnam-si 11962, Gyeonggi-do, Korea; orthopaedee@naver.com; 2Paley Orthopedic and Spine Institute, Kimmel, 901 45th St, West Palm Beach, FL 33407, USA

**Keywords:** hip, multiple hereditary exostoses (MHE), femuro-acetabular impingement, coxa valga, surgical hip dislocation, varus intertrochanteric osteotomy

## Abstract

The hip joint involvement in multiple hereditary exostoses (MHE) occurs in 30–90%, causing pain and limitation of motion by femoroacetabular impingement, coxa valga, acetabular dysplasia, hip joint subluxation, and osteoarthritis. The purpose of this study was to investigate the clinical and radiographic outcomes of ten hips in seven patients treated by surgical dislocation and corrective osteotomies between 2004 and 2009. Surgical dislocation and excision of the osteochondromas and varus intertrochanteric osteotomies were performed in all cases when the neck–shaft angle was > 150°. Common sites of osteochondromas were medial, posterior, and anterior neck of the femur. Neck–shaft angle of the femur was improved from a mean of 157° to 139°, postoperatively. On an average, the center-edge angle improved from 20° to 30° postoperatively. We believe that Ganz’s safe surgical dislocation technique is the preferred treatment of MHE. This safeguards the circulation of the femoral head and the osteochondromas can be resected under direct vision. It can be combined with additional corrective osteotomies because the hip affected by MHE is frequently associated with dysplastic changes which can result in premature osteoarthritis.

## 1. Introduction

Multiple hereditary exostoses (MHE) is an autosomal-dominant disorder with a prevalence of 1:50,000 persons within the general population [1]. The EXT1 and EXT2 genes located at 8q24 and 11p11–p12, respectively, are known to be associated with MHE [2,3].

A variety of problems can be related to MHE and the majority of these problems are related to the bony protrusions affecting the surrounding joints, muscles, tendons, nerves, blood vessels, and skin. Common problems are pain, restricted range of motion, and cosmetic concerns. Pain may come from repeated motion over prominent osteochondroma, bursa formation, and joint impingement [1,4]. Some osteochondromas tether the growth plate. This can lead to asymmetric growth of the growth plate and consequent limb deformity. The common deformities of the lower extremity include short stature, leg length discrepancy, and valgus deformities of the knee, ankle, and proximal femur [4,5].

MHE incidence at the proximal femur and pelvis has been reported from 30% to 90% and 15% to 62%, respectively [1,6,7]. The hip area involvement can lead to coxa valga [5,8,9,10,11], acetabular dysplasia [5,9], femoroacetabular impingement (FAI) [9,10,12], hip joint subluxation [9,10,12], and osteoarthritis [13,14]. Major concerns in the hip joint are pain and limited range of motion by FAI. Dysplastic hip which can be caused by FAI and by coxa valga is also a big problem [9,10,11]. Thus, proper treatment in MHE hip is important to eliminate pain and limit motion to prevent premature hip arthritis. There have been only a few reports about the surgical treatment of one- or two-hip case series of hip exostoses due to MHE [10,11,12,15,16,17]. The purpose of this study was to investigate the clinical and radiographic outcomes of ten hips in seven patients treated by surgical dislocation and corrective osteotomies.

## 2. Material and Methods

All patients who underwent hip surgery due to MHE were included, and patients who did not follow-up after surgery or did not have appropriate X-rays or medical records were excluded from the study. The records of ten hips in seven patients with MHE treated by surgical dislocation of the hip joint between 2004 and 2009 were retrospectively reviewed (Table 1). The main reasons for visiting the clinic were hip pain (ten hips) and motion limitation (nine hips). The hip pain was aggravated with motion especially flexion, internal rotation, and adduction. One case presented a sciatica-like symptom of the right leg (Figure 1a–c). The diagnosis was initially made on plain radiographs. One patient was diagnosed with Langer–Giedion syndrome [18], which had MHE with a unicameral cyst (Figure 2a,b). Three-dimensional reconstructive computerized tomography and bone scanning were performed in selected persons for accurate surgical planning if the location and the shape of the mass was not clearly identified by plain x-ray. The study included four male and three female patients, and the average age at the time of surgery was 22 (range, 6–39) years. The right side was involved in one patient, the left side in one patient, and both sides in five patients. However, two patients underwent surgery on the unilateral side because the other side was free from any disability among the patients with both lesions. 

The radiographic assessment of the proximal femur was performed by evaluating the location of the mass, the femoral neck–shaft angle, Shenton’s line, and the ratio of the diameter of the femoral neck to that of the shaft (NSR) as an index of femoral neck overgrowth and surgical excision amount [1,5]. Moreover, NSR was measured as the diameter of the femoral neck divided by the diameter of the femoral shaft just distal to the lesser trochanter (Figure 3a–d). The acetabulum was evaluated by Sharp’s acetabular angle and the center-edge angle of Wieberg [19,20].

All ten hips underwent surgery to excise osteochondromas using safe surgical hip dislocation [21] by the senior author (DP). One female and two male patients had staged bilateral surgeries. Varus intertrochanteric osteotomy was performed when the neck–shaft angle was more than 150° (Figure 4a,b). After surgery, follow-up was performed at 4-week intervals until a bony union was established, and thereafter, observations were made at intervals of 6 months to 1 year. All of the patients were followed with serial radiographs for a mean follow-up of 49 (range, 12–97) months. 

### 2.1. Surgical Procedure

In most of the cases, surgical dislocation and excision of the osteochondromas were performed with the use of surgical dislocation of the hip described by Ganz et al. [21]. A Kocher–Langenbeck incision and a greater trochanter flip osteotomy were performed with the patient in a lateral position on a radiolucent table. A trochanteric osteotomy was performed with a thickness of 1.5 cm along the line of the posterior margin of the vastus lateralis just anterior to the most posterior insertion of the vastus medius. This left the posterior 2/3rd of the piriformis tendon and all the other short external rotators, including the obturator externus, intact and preserved the deep branch of the medial femoral circumflex artery. The capsule can be exposed through the interval between the gluteus minimus and the tendon of piriformis with the trochanteric fragment including the gluteus minimus flipped anteriorly. The z-shaped capsulotomy was made carefully not injuring the labrum. The hip was anteriorly dislocated, which exposes the whole femoral head and acetabulum with leg manipulation. Removal of the osteochondromas was performed with osteotomes and a small high-speed burr. Meticulous determination of the extension of osteochondroplasty was made with a spherometer gage (Wright Medical Technology, Arlington, TN, USA; Figure 3). The acetabulum was then examined to check the status of the cartilage and labrum. The hip was relocated and put through a full range of motion to make sure that the impingement was eliminated. A 3.5 mm cortical or cannulated screw was inserted along the axis of the femoral neck to prevent incidental fracture of the femoral neck (Figure 3). The greater trochanter was reattached with three 3.5 mm cortical screws. The proximal fragment was fixed by the blade plate when varus intertrochanteric osteotomy was undertaken.

### 2.2. Postoperative Management

Patients were allowed to touch down weight-bearing postoperatively using bilateral crutches in an abduction brace for 6 weeks.

### 2.3. Ethical Considerations

All participants provided informed consent for participation. This article does not disclose any personally identifiable data of any of the participants in any form. Thus, consent for publication is not applicable here. All study-related procedures were conducted following the rules of the 1975 Declaration of Helsinki.

## 3. Results

### 3.1. Clinical Results

The pain was completely resolved with the excision of the mass in all cases. One patient had a sciatica-like symptom that was suspected to be compression neuropathy. After surgical decompression with the excision of osteochondromas, the symptoms of the patient were completely resolved (Figure 2). The range of motion of the hip was improved as follows: flexion from a mean of 65° (range, 10°–110°) preoperatively to a mean of 104° (range, 90°–130°); extension from −3° (range, −20°–0°) to 0°; adduction from 12° (range, 0°–30°) to 26° (range, 15°–40°); abduction from 30° (range, 0°–60°) to 44° (range, 20°–60°); internal rotation from 21° (range, 0°–45°) to 30° (range, 15°–45°); and external rotation from 31° (range, 10°–60°) to 32° (range, 10°–50°) (Table 2).

### 3.2. Radiographic Results

Osteochondromas were noted around the femoral neck in nine cases. The medial side of the femoral neck was involved in all patients, posterior in nine of ten hips, and anterior in eight. Osteochondromas involved the femoral head in the Langer–Giedion syndrome (Figure 1). NSR in mediolateral width was improved with surgery from a mean of 3.3 (range, 2.1–6.7) to 1.8 (range, 1.0–3.0) and in the anteroposterior width from 2.7 (range, 1.9–4.3) to 1.4 (range, 0.8–2.1). Neck–shaft angle was improved from a mean of 157° (range, 135°–180°) to 139° (range, 130°–150°); and center-edge angle from 20° (range, 0°–30°) to 30° (range, 25°–40°). Sharp’s acetabular angle was 44° on average (range, 40°–45°) preoperatively. Shenton’s line was broken in six hips and postoperatively reduced in all cases (Table 3).

### 3.3. Combined Surgery and Complications

All the additional correctional osteotomies were performed during the index surgery. Varus intertrochanteric osteotomy was done in five hips. Valgus intertrochanteric and distal femoral varus osteotomies were performed in one case (Langer–Giedion syndrome). However, no pelvic osteotomy was done.

None of the patients has undergone additional reconstructive hip surgery at a mean of 4.1 (range, 1.0–8.1) years postoperatively. Avascular necrosis of the femoral head, fracture of the femoral neck, fracture of the greater trochanter, and nonunion were not observed.

## 4. Discussion

The main problems with osteochondromas of the femoral head and neck were FAI pain anteriorly and/or posteriorly depending on the location of the exostoses and the limitation of the hip motion. In this series, all the hips presented both of these problems. FAI is a well-known cause of hip pain, limitation of motion, and labral/chondral abnormalities [22,23,24]. Ganz et al. [25] proposed that FAI is a mechanism for the development of early arthritis for most nondysplastic hips. Moreover, they explained the pathomechanism of FAI as dynamic abnormal contact between the femoral head–neck junction and the acetabular rim resulting in damage to the acetabular cartilage and labrum [25]. Consequently, FAI can be managed by open dislocation or arthroscopic surgery [21,22,23,24,26,27,28]. Ganz et al. [21] developed a safe technique to dislocate the hip joint without risk of avascular necrosis (AVN) of the femoral head. The potential risk of the surgical dislocation includes AVN of the femoral head, femoral neck fracture, and trochanteric nonunion or bursitis. Beaulé et al. [23] observed 26% of trochanteric bursitis and 3% of trochanteric nonunion among 37 hips. Furthermore, the safety of this approach has been proved by several clinical reports, and no AVN has been reported by this technique [11,17,22,23]. Arthroscopic surgery has recently become a promising alternative for FAI [24,26,29,30,31,32,33]. Arthroscopic treatment is a less invasive procedure to allow early rehabilitation and theoretically has a lower risk for AVN. It could be an attractive alternative especially in active young patients with FAI [29,33]. However, the arthroscopic approach has definite limitations in the treatment of MHE hip. It has a difficulty in visualizing posterior femoral lesions and is inappropriate to treat combined deformity of the extremity. In the author’s series, posterior femoral lesions exist in 90% of the cases. Moreover, a concomitant femoral osteotomy was performed in 60% of the cases. This study has not experienced any possible complications of open surgical dislocation and we also prefer to insert a cortical screw for preventing femoral neck fracture after heavy mass resection.

Several hip deformities have been documented including femoral anteversion, coxa valga, acetabular dysplasia, and hip subluxation [4,5,9,10,34,35]. Lesion incidence of the proximal femur has been reported as 30–90%, and of the coxa valga as 25% [1,6,7,34,35]. A true developmental hip dysplasia may result from mechanical insufficiency of the acetabular cavity due to osteochondromas of the proximal femur and from coxa valga [10,36]. Consequently, subluxation of the hip occurs due to the effect of the osteochondroma pushing the hip out of the joint combined with the effect of the valgus of the femoral neck [9,10]. Complete dislocation of the hip can even occur, as presented in this series (Figure 1). Furthermore, severe MHE can result in total hip replacement [14]. Additional corrective osteotomy should be considered at index surgery to restore adequate congruency of the hip joint to prevent this deteriorative course. Six hips (60%) were necessary to undergo additional corrective osteotomies in this series.

Eight reports have been published regarding surgical treatment of MHE hip [10,11,12,15,16,17,37,38]. Sorel et al. [37] reported 20 symptomatic osteochondromas of the femoral neck. They observed that the mean range of motion significantly increased in all directions and postoperatively; the pain associated with the lesion either considerably decreased or was non-existent. Although some complications were observed in four patients, including pseudoarthrosis of the trochanteric osteotomy, traumatic separation of the trochanteric osteotomy, a peritrochanteric femoral fracture, and AVN, they concluded that Ganz’s surgical dislocation provided a reliable approach to osteochondromas of the femoral neck and offered noteworthy improvement in the quality of life, including pain and range of motion of the hip. Shin et al. [17] reported 23 pediatric hip disease cases treated using Ganz’s surgical hip dislocation technique. Nine hips with MHE were included in this series, and all nine MHE hips showed FAI due to sessile osteochondromas of the femoral neck. They were treated by osteochondroplasty via a surgical hip dislocation approach. In one patient, femoral varization and derotation osteotomy were combined. They observed that the range of hip flexion improved from a preoperative value of 82.8° (range, 60°–110°) to a postoperative value of 108.9° (range, 90°–130°) and that there was no increase in hip pain or stiffness after the surgical hip dislocation. They observed one case of postoperative AVN of the femoral head in unstable slipped capital femoral epiphysis. It was highly likely that the delayed intervention caused AVN rather than the surgical hip dislocation approach itself, given that surgical intervention was delayed by 1 week for medical reasons, and the femoral head was found to be avascular intraoperatively. However, no detailed clinical and radiographic description of MHE cases was found. Among the four reports of the two-hip case series, Jellicoe et al. [11] gained completely asymptomatic hips with 2 and 3 years of follow-up using Ganz surgical dislocation. However, they observed a still-subluxed hip at the last follow-up and felt that additional surgery may be needed. Felix et al. [9] reported an MHE patient with bilateral hip subluxation and acetabular dysplasia. Their treatment included bilateral femoral varus derotation osteotomies through posterior approach and bilateral acetabular osteotomies (steel pelvic osteotomy) using an adductor approach. They observed an asymptomatic patient and obtained a good range of motion at 2 years of follow-up. Moreover, Woodward et al. [12] reported two cases of children with MHE in whom painful restriction of hip movement developed due to intra-acetabular osteochondromata. Excision of the lesions relieved pain and restored joint movement after 14 and 3 months of follow-up, respectively. The other two case reports by Bonnomet et al. [16] described the cases of two children with MHE hip. The lesions were primarily located in the acetabular fossa and caused pain and limitation of range of motion. The exostoses were removed using hip arthroscopy, which is a less invasive approach. Although they obtained good results in terms of pain relief and range of motion at 3 years of follow-up, they did not describe deformities and dysplastic changes of the hip which is one of the mainstream MHE treatments. These cases constitute a new and interesting application of hip arthroscopy. In a single-hip case report, Ofiram and Porat [10] reported a case of intra-articular and extra-articular osteochondromas in the right hip which caused hip subluxation. They performed arthrotomy using Smith–Peterson approach without a concomitant osteotomy in the presence of severe acetabular dysplasia. However, they opened the possibility of second-stage osteotomies. It is believed that the author’s series on the surgical hip treatment in MHE is one of the largest and the most detailed one.

The limitation of this paper is that the number of cases is not large enough to give statistical significance, and data on clinical indicators for evaluating pain or quality of life are not presented. In addition, the results of long-term follow-up are expected to be necessary in the future.

According to the author’s experience, Ganz’s safe surgical dislocation is the treatment of choice for the MHE hip. This safeguards the circulation of the femoral head avoiding AVN of the femoral head. However, good knowledge of the anatomy of the vasculature around the hip joint is essential. The osteochondromas can be resected under direct vision and the femoral head templated with a spherical template to ascertain if the femoral head is spherical. Moreover, it can be combined with a varus osteotomy using a blade plate for fixation. Furthermore, hips affected by MHE are frequently associated with dysplastic changes and should be carefully evaluated for the necessity of additional osteotomy to restore proper coverage and containment.

## Figures and Tables

**Figure 1 children-08-00490-f001:**
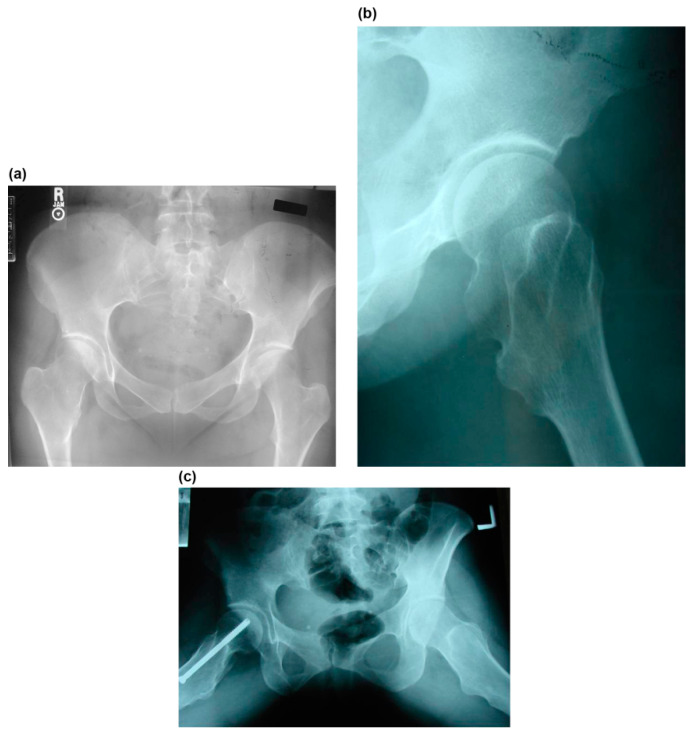
A 14-year-old girl complained of posterior thigh pain at the right lower extremity that was aggravated with straight-leg raising (case no. 2). (**a**) Preoperative anteroposterior radiograph of the right hip shows a small bump at the medial side of the femoral neck. (**b**) Preoperative lateral radiograph of the right hip shows a large bump at the posterior side of the femoral neck. (**c**) The sciatica-like symptom was completely resolved after complete resection of the osteochondromas of the posterior femoral neck.

**Figure 2 children-08-00490-f002:**
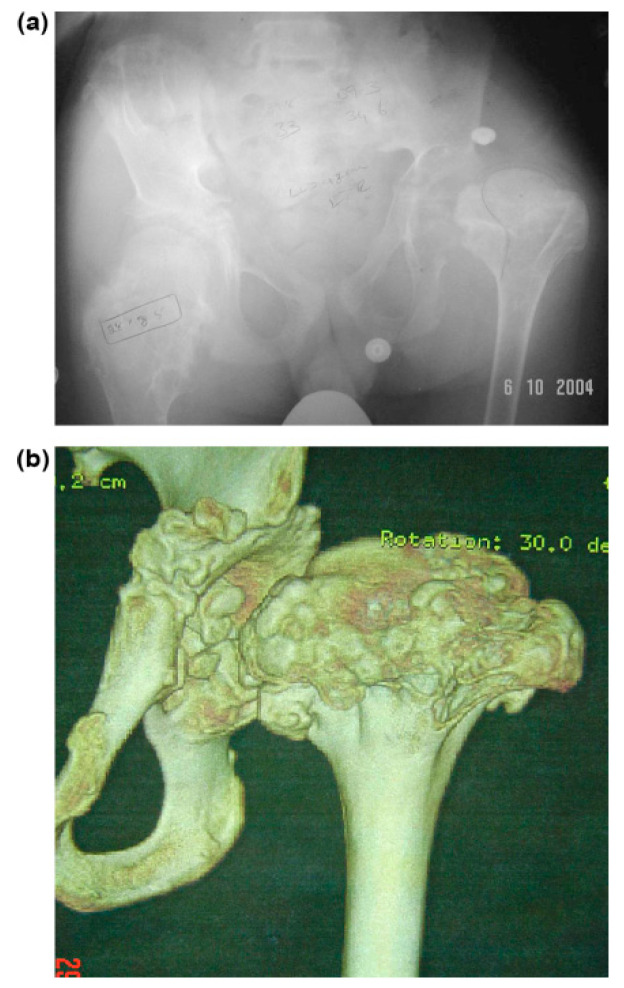
The images illustrate the case of a 14-year-old boy with Langer–Giedion syndrome (case no. 3). (**a**) Preoperative anteroposterior radiograph of the left hip shows osteochondromas at the femoral head and a completely dislocated hip. (**b**) Dislocated hip.

**Figure 3 children-08-00490-f003:**
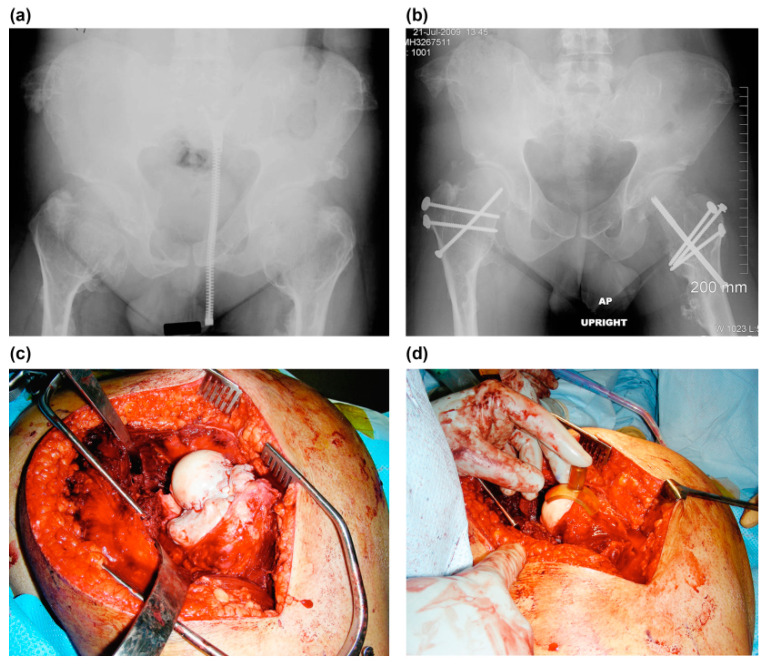
The images show the case of a 23-year-old man with symptomatic bilateral femoroacetabular impingement pain and limitation of motion (case no. 5). (**a**) Preoperative anteroposterior radiograph shows a large bump at the medial side of the femoral neck. Neck–shaft angle of the femur was within tolerable range (Rt, 140; Lt, 150). The containment of the hip was normal. (**b**) Postoperative anteroposterior radiograph of the hip. Mediolateral neck–shaft ratio improved from 6.7 to 1.6 on the right hip, 3.6 to 1.7 on the left hip. A cortical or cannulated screw was inserted through the femoral neck to prevent femoral neck fracture and the osteotomized greater trochanter was fixed with two-three 3.5 mm cortical screws. (**c**) Intraoperative photograph of the left hip. (**d**) The osteochondroplasty of the femoral neck can be performed with small osteotomes and a high-speed burr. When the resection is satisfactory, a special template (spherometer gage) can be used to check the sphericity of the remaining femoral head. The neck–shaft ratio improved from 3.6 to 1.7 mediolaterally and 2.7 to 1.3 anteroposteriorly.

**Figure 4 children-08-00490-f004:**
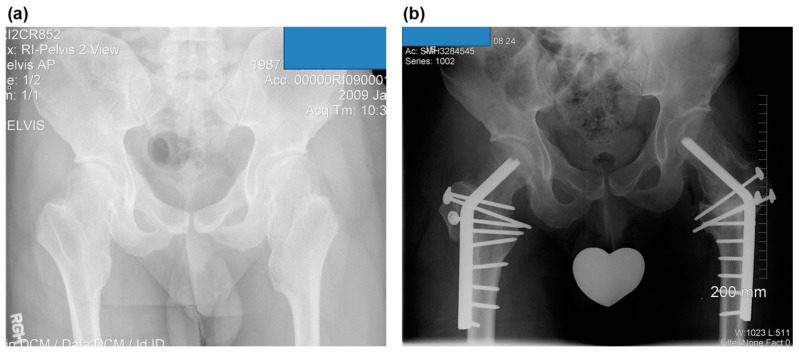
The case of a 22-year-old man with symptomatic bilateral femoroacetabular impingement in MHE (case no. 6). (**a**) A preoperative anteroposterior radiograph of the hip shows a large bump at the medial side of the femoral neck and severe coxa valga. (**b**) Postoperative anteroposterior radiograph of the hip. Valgus intertrochanteric osteotomies were performed on both hips using blade plates, and the trochanteric osteotomy sites were fixed with 3.5 mm cortical screws.

**Table 1 children-08-00490-t001:** Demographic data.

Case	Gender	Involved Side	Age at Operation (Years)	Follow-Up Period (Months)	Site of Tumor at Hip	Symptoms at Presentation
1	Male	R	6	84	A/M/P	Hip pain, LOM
2	Female	R	39	80	M/P	Sciatic nerve Sx., Hip pain
3	Male	L	14	97	FH	Hip pain, LOM
4	Female	L	11		M/P	Hip pain, LOM
5	Male	B	R: 23	50	A/M/P	Hip pain, LOM
			L: 26	13	A/M/P	Hip pain, LOM
6	Male	B	R: 22 L: 25 R: 26	12	A/M/P	Hip pain, LOM
7	Female	B	L: 22	19	A/M/P	Hip pain, LOM
			R: 28		A/M	Hip pain, LOM
			L: 27	40	A/M/P	Hip pain, LOM

A, anterior; M, medial; P, posterior; R, right; L, left; B, both; LOM, limitation of motion; FH, femoral hip.

**Table 2 children-08-00490-t002:** Changes in range of motion of the hip joint.

Case		Range of Motion (°)												Additional Procedure
		Flexion		Extension		Adduction		Abduction		IR		ER		
		Pre	Post	Pre	Post	Pre	Post	Pre	Post	Pre	Post	Pre	Post	
1		10	110	−10	0	5	20	45	45	10	35	60	50	VIO
2		110	110	0	0	20	20	45	45	20	20	40	20	Sciatic nerve decompression
3		60	90	0	0	0	20	0	20	0	20	20	20	Langer–Giedion syndrome VaIO; DFVO
4		60	130	0	0	0	20	60	60	10	20	40	40	VIO
5	R	60	100	−20	0	0	15	30	60	0	15	20	40	
	L	60	105	0	0	5	20	20	60	20	20	20	30	
6	R	70	90	0	0	15	30	30	30	20	30	45	45	VIO
	L	70	100	0	0	20	30	30	30	45	45	10	10	VIO
7	R	90	100	0	0	20	40	20	45	40	45	20	30	VIO
	L	60	100	0	0	30	40	20	45	45	45	30	30	

R, right; L, left; VIO, varus intertrochanteric osteotomy; VaIO, valgus intertrochanteric osteotomy; DFVO, distal femoral varus osteotomy; Pre, pre-operation; Post, post-operation.

**Table 3 children-08-00490-t003:** Changes in radiographic parameters.

Case		NSA (°)		NSR				SL		SAA (°)	CEA (°)	
		Preoperative	Postoperative	AP		Lateral		Preoperative	Postoperative		+82-	Posterative
				Pre	Post	Pre	Post					
1		180	135	3.6	2.4	3.3	2.1	Broken (sublux.)	Reduced	45	0	25
2		145	145	2.1	1.2	1.9	0.8	Intact	Intact	45	30	30
3		135	135	5.0	3.0	4.3	1.9	Broken (dislocation)	Reduced	40	<0	30
4		155	140	2.5	1.7	1.9	1.3	Broken (sublux.)	Reduced	45	15	25
5	R	140	140	6.7	1.6	2.3	1.2	Intact	Intact	45	25	30
	L	150	150	3.6	1.7	2.7	1.3	Intact	Intact	45	25	30
6	R	170	140	2.3	1.6	2.9	1.4	Broken (sublux.)	Reduced	45	25	40
	L	180	130	2.3	2.0	2.5	1.3	Broken (sublux.)	Reduced	45	20	30
7	R	160	140	2.7	1.7	2.5	1.5	Broken (sublux.)	Reduced	40	10	25
	L	150	135	2.1	1.0	2.1	1.5	Intact	Intact	40	30	30

R, right; L, left; NSA, neck–shaft angle; NSR, neck–shaft ratio; SL, Shenton’s line; SAA, Sharp’s acetabular angle; CEA, center-edge angle of Wieberg; Pre, pre-operation; Post, post-operation.

## Data Availability

The datasets generated during and/or analyzed during the current study are available from the corresponding author on reasonable request.

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
