# Peer review of "Reconstruction of the Hip in Multiple Hereditary Exostoses"

_children, 2021, doi:10.3390/children8060490_

Round 1
Reviewer 1 Report
The manuscript is a retrospective study aimed to investigate the clinical and radiographic outcomes of 10 hips in seven patients treated by surgical dislocation and corrective osteotomies between 2004 and 2009. Surgical dislocation and excision of the osteochondromas and varus intertrochanteric osteotomy were performed in all cases when the neck–shaft angle was >150°. Neck–shaft angle of the femur was improved from a mean of 157° to 139°, postoperatively. On an average, center edge angle improved from 20° to 30° postoperatively. The authors believe that Ganz’s safe surgical dislocation is the preferred treatment in MHE.
I read the article with interest, the title is well thought out and faithfully reflects the content of the study.
- The abstract is sufficiently developed, and it is useful to characteristic of the study, but a few concerns are present:
Comment 1: The purpose of the retrospective study should be included in the abstract for a better understanding of the study.
Comment 2: In line 2 there is a parenthesis that appears to have not been opened before: "The hip joint involvement in multiple hereditary exostoses (MHE) is 30%–90%, causing 9 pain and limitation of motion by femoroacetabular impingement)," Please correct this mistake.
- In the introduction, the characteristics of supracondylar humerus fracture in the pediatric population Hereditary multiple exostoses have been accurately described, even if a little too synthetic.
Comment 3: Some references should be added regarding the diagnosis, treatment and prognosis that can occur after this type of patients, for example: (Pacifici M. (2017) "Hereditary Multiple Exostoses: New Insights into Pathogenesis, Clinical Complications, and Potential Treatments").
Comment 4: Also in the introduction, the purpose of the retrospective study should be included for a better understanding to the reader.
- In materials and methods, the surgical procedure and postoperative management have been adequately developed.
Comment 5: " Three-dimensional reconstructive computerized tomography and bone scanning were performed in selected persons when necessary for surgical planning." What kind of criteria did you use for your chosen people?
Comment 6: The criteria for inclusion and exclusion of patients in the study do not seem to be very clear, please clarify this aspect
Comment 7: "All of the patients were followed with serial radiographs for a mean follow-up of 49 (range, 12–97) months." How often did you perform X-ray checks on your patients?
Comment 8: "Patients were allowed to touch down weight-bearing postoperatively using bilateral crutches in an abduction brace for 6 weeks." after 6 weeks of bilateral crutches in an abduction brace, what type of rehabilitation protocol was performed? Was the use of analgesics necessary? and with what average frequency?
- The discussion is sufficiently developed, even if a little too synthetic.
Comment 9: The limits of the study do not seem to be specify, it would be appropriate to refer to them.
Finally, English language editing is needed.
Author Response
We appreciate the time and effort that you have dedicated to providing your valuable feedback on my manuscript. We are grateful to the reviewers for their insightful comments on my paper. We have been able to incorporate changes to reflect most of the suggestions provided by the reviewers. The changes can be seen in tracks in the uploaded manuscript
Here is a point-by-point response to the reviewers’ comments and concerns. Comments from Reviewer
- The abstract is sufficiently developed, and it is useful to characteristic of the study, but a few concerns are present:
Comment 1: The purpose of the retrospective study should be included in the abstract for a better understanding of the study.
Author Response: Thank you for the comment. I changed the sentence in line 3 of ABSTRACT to 'The purpose of this study is to investigate...'
Comment 2: In line 2 there is a parenthesis that appears to have not been opened before: "The hip joint involvement in multiple hereditary exostoses (MHE) is 30%–90%, causing 9 pain and limitation of motion by femoroacetabular impingement)," Please correct this mistake.
Author Response: Correction was done
- In the introduction, the characteristics of supracondylar humerus fracture in the pediatric population Hereditary multiple exostoses have been accurately described, even if a little too synthetic.
Comment 3: Some references should be added regarding the diagnosis, treatment and prognosis that can occur after this type of patients, for example: (Pacifici M. (2017) "Hereditary Multiple Exostoses: New Insights into Pathogenesis, Clinical Complications, and Potential Treatments").
Author Response: The reference you mentioned was added as reference number 38.
Comment 4: Also in the introduction, the purpose of the retrospective study should be included for a better understanding to the reader.
Author Response: I changed the sentence to 'The purpose of this study is to investigate...' in Introduction as well.
- In materials and methods, the surgical procedure and postoperative management have been adequately developed.
Comment 5: " Three-dimensional reconstructive computerized tomography and bone scanning were performed in selected persons when necessary for surgical planning." What kind of criteria did you use for your chosen people?
Author Response: I corrected the sentence to ‘Three-dimensional reconstructive computerized tomography and bone scanning were performed in selected persons for accurate surgical planning for surgical planning if the location and the shape of the mass is not clearly identified by plain x-ray.’
Comment 6: The criteria for inclusion and exclusion of patients in the study do not seem to be very clear, please clarify this aspect
Author Response: I added ‘ All patients who underwent hip surgery due to HME were included, and patients who did not follow-up after surgery or did not have appropriate X-rays or medical records were excluded from the study.’
Comment 7: "All of the patients were followed with serial radiographs for a mean follow-up of 49 (range, 12–97) months." How often did you perform X-ray checks on your patients?
Author Response: I added ‘After surgery, follow-up was performed at 4-week intervals until a bony union was established, and thereafter, observations were made at intervals of 6 months - 1 year.’
Comment 8: "Patients were allowed to touch down weight-bearing postoperatively using bilateral crutches in an abduction brace for 6 weeks." after 6 weeks of bilateral crutches in an abduction brace, what type of rehabilitation protocol was performed? Was the use of analgesics necessary? and with what average frequency?
- The discussion is sufficiently developed, even if a little too synthetic.
Comment 9: The limits of the study do not seem to be specify, it would be appropriate to refer to them.
Author Response: I added ‘The limitation of this paper is that the number of cases is still small enough to give statistical significance, and data on clinical indicators for evaluating pain or quality of life are not presented. In addition, the results of long-term follow-up are expected to be necessary in the future.’
Reviewer 2 Report
Thank you for sharing your results. It would be instructive if you could add objective functional results for these patients, if available. Please also consider adding more detail in the methods section for the varus osteotomy given the number of patients who required this procedure.
Author Response
We appreciate the time and effort that you have dedicated to providing your valuable feedback on my manuscript. We are grateful to the reviewers for their insightful comments on my paper. We have been able to incorporate changes to reflect most of the suggestions provided by the reviewers. The changes can be seen in tracks in the uploaded manuscript
Here is a point-by-point response to the reviewers’ comments and concerns. Comments from Reviewer
Comments and Suggestions for Authors
- Thank you for sharing your results. It would be instructive if you could add objective functional results for these patients, if available. Please also consider adding more detail in the methods section for the varus osteotomy given the number of patients who required this procedure.
Author Response: The functional score was not included in the results of this study, and this was described as a limitation of the study. I would appreciate if you understand that detailed descriptions for the varus osteotomy was omitted since it has been described in many other papers and it is not the focus of this paper .